# Changes in Carbon and Nitrogen Metabolites before, at, and after Anthesis for Wheat Cultivars in Response to Reduced Soil Water and Zinc Foliar Application

**DOI:** 10.3390/plants11091261

**Published:** 2022-05-06

**Authors:** Rasha E. Mahdy, Sameera A. Alghamdi, Ahmed Amro, Suzan A. Tammam

**Affiliations:** 1Agronomy Department, Faculty of Agriculture, Assiut University, Assiut 71515, Egypt; 2Department of Biological Sciences, Faculty of Science, King Abdulaziz University, Jeddah 21589, Saudi Arabia; saalghamdy1@kau.edu.sa; 3Department of Botany and Microbiology, Faculty of Science, Assiut University, Assiut 71515, Egypt; ahmed.mohamed20@science.aun.edu.eg (A.A.); or susan@aun.edu.eg (S.A.T.)

**Keywords:** *Triticum aestivum*, Zn, water stress, amino acids, soluble proteins, soluble sugars

## Abstract

Water deficit stress is one of the major constraints for commercial agriculture, as it disturbs the metabolic processes in plant. Identification of carbon and nitrogen receptors that act on drought resistance helps in breeding for drought resistance varieties. Zn fertilizer can regulate multiple antioxidant defense systems at the transcriptional level in response to drought. Two field experiments were conducted in 2018–2019 and 2019–2020 seasons to explore the effectiveness of foliar application of zinc oxide on soluble sugar, soluble proteins, and free amino acids under normal irrigation and drought-stressed environments. Three Egyptian wheat cultivars (*Triticum aestivum* L.) were used. The experimental design was split-plot in RCBD with three replications, applying zinc oxide levels to the whole plot and the split plots. Leaf samples were taken for analysis before anthesis, at anthesis, and after anthesis. Application of Zn increased soluble sugars. However, the free amino acids were higher under irrigation, reached the maximum at anthesis, and decreased sharply after 2 weeks from anthesis. The ranking of cultivars for the three metabolites differed according to plant stage, reflecting the response to Zn and years. Correlations between metabolites according to Zn were positive. The findings suggest the potential of foliar application of Zn to alleviate drought stress.

## 1. Introduction

A prolonged period of deficient precipitation or irrigation is one of the most challenging problems in crop production worldwide. Drought stress adversely affects nutrient uptake, which results in a reduction in plant growth, yield, and grain quality [1,2]. Furthermore, water deficit can cause negative reversible and irreversible physiological changes to plant state in the vegetative and reproductive periods of plant development [3,4].

Wheat (*Triticum aestivum* L.) is an extensively used crop worldwide. It is well documented that cereal crops, especially wheat, are susceptible to drought and Zn deficiency. Under severe water deficit, the soluble protein decreased, and the free amino acids, proline, total soluble sugars (TSS), and fructose increased. Such an increase was more pronounced during tillering than during jointing and differed according to genotype [5]. After irrigation, the concentrations of (TSS) and fructose decreased, and rate of recovery was lower under severe than moderate stress treatments [6]. Drought stress probably decreased the grain protein content and grain yield [7]. On the other hand, water deficit in the stem and leaves reduced the level of carbohydrate reserves involved in grain filling, causing a reduction in seed mass [8]. Hence, drought stress had an inhibitory effect on relative leaf water and chlorophyll content, while markedly increasing the proline and sugar contents of leaves in wheat [9,10].

Micronutrients play an important role in plant growth and development. Fe and Zn applications, whether soil or foliar application, increased the grain yield, as well as the protein and gluten content, of wheat [11]. High pH and other factors in the soil fixed Zn, caused Zn deficiency, and reduced amino-acid accumulation in plant tissues and protein synthesis [12]. Zinc also is essential in tryptophan synthesis, cell division, photosynthetic maintenance, and membrane structure, and it acts as an essential cofactor in controlling the biosynthesis of proteins [13,14,15,16]. Zinc fertilization enhances cereal, vegetable, and food production. On the other hand, Zn deficiency reduces leaf size, leaf area, chlorophyll, cross-sectional area, yield, and kernel size, and it causes interveinal necrosis and ribbed leaf margins [17]. Under the extreme conditions of zinc shortage, catalase, superoxide dismutase, and glutathione peroxide activities often increase [18].

Apparently, Zn plays a role in alleviating wheat plant drought stress through the Zn-mediated increase in photosynthesis pigments and active oxygen scavenging substances, as well as a reduction in lipid peroxidation. Furthermore, Zn fertilizer could regulate multiple antioxidant defense systems at the transcriptional level in response to drought [19]. The aim of this work was to study the effect of foliar application of Zn oxide on soluble sugar, proteins, and amino acids 2 weeks before anthesis, at anthesis, and 2 weeks after anthesis of wheat (*Triticum aestivum* L.), as well as on the alleviation of drought stress and the correlations between various primary metabolites of wheat cultivars under the effects of Zn application.

## 2. Results

### 2.1. Variances

#### 2.1.1. Soluble Sugars

The mean squares of years were significant (*p* ≤ 0.01) in the three plant stages (2 weeks before anthesis, at anthesis, and 2 weeks after anthesis) for the three metabolites, indicating the effects of climatic factors (Table 1). The effects of Zn on soluble sugars were significant (*p* ≤ 0.01) in both years except in the first year, under drought at the last stage, and the first year under irrigation at 2 weeks before anthesis. The three cultivars differed significantly (*p* ≤ 0.05 or *p* ≤ 0.01) under both environments except in the second year at anthesis under drought. Both Zn levels and cultivars interacted significantly with years, and the cultivars interacted significantly with Zn levels.

#### 2.1.2. Soluble Proteins

The Zn levels varied significantly under both environments except in the combined under drought at 2 weeks after anthesis and in the second year under irrigation. The Zn levels interacted significantly (*p* ≤ 0.01) with years (Table 1). The three cultivars varied significantly in terms of soluble proteins under both environments except at anthesis. Except for the anthesis stage, the first-order interaction of cultivars with Zn and years indicated the differential response of cultivars to foliar application of Zn from year to year.

#### 2.1.3. Free Amino Acids

The Zn levels significantly (*p* ≤ 0.05 or *p* ≤ 0.01) affected free amino acids under drought except in the first year at 2 weeks before anthesis (Table 1). However, under irrigation, free amino acids were not affected by Zn levels in the 2 years at 2 weeks before and after anthesis. The three cultivars under drought did not differ in terms of free amino acids except in the second year at anthesis and the first year 2 weeks after anthesis. Otherwise, under irrigation, the cultivars significantly (*p* ≤ 0.01) differed in terms of amino acids at all plant stages except in the second year at 2 weeks after anthesis. The interaction of Zn levels with years was significant (*p* ≤ 0.01) in four out of the six cases in both environments. Likewise, the cultivars showed a differential response to Zn levels in most cases.

### 2.2. Means of Zn Levels

Means of the three metabolites were larger in the first than in the second year (Table 2) for the four levels of Zn except for a few cases of free amino acids under irrigation, confirming the significant (*p* ≤ 0.01) effect of years (Table 1). The soluble sugars mostly increased from 2 weeks before anthesis to 2 weeks after anthesis for the four levels of Zn (Table 1 and Figure 1). The Zn levels 500 and 250 ppm were the best at increasing soluble sugars under both environments. Otherwise, the concentration of soluble proteins was higher in the control treatment than at 500 and 250 ppm of Zn O. Comparing the soluble sugars and proteins of treatment ‘none’ under drought with those under irrigation, it can be noted that drought stress increased both metabolites. However, the free amino acids decreased under drought (Table 2, Figure 1, Figure 2 and Figure 3).

Regarding the effect of zinc on free amino acids (Table 2, Figure 3), it can be noted that its concentration was higher under irrigation than under drought stress, and it reached its maximum at anthesis, before decreasing sharply 2 weeks after anthesis, whether under irrigation or drought stress. The zinc levels had no clear effect on free amino acids.

### 2.3. Means of Cultivars

Means of cultivars in terms of both soluble sugars and proteins according to Zn levels and years were higher under drought than under normal irrigation (Figure 4 and Figure 5). The means increased from 2 weeks before anthesis to 2 weeks after anthesis under both environments. However, the ranking of cultivars differed according to plant stage under both environments, reflecting the significant (*p* ≤ 0.01) second-order interaction of Zn × cultivar × years (Table 1). The free amino acids behaved the same except that the highest concentrations were recorded at the anthesis stage (Figure 6).

## 3. Correlation between Different Metabolites in Wheat Cultivars under the Effect of Various Zinc Concentrations

Apparently, all significant correlations between investigated metabolites of wheat cultivars were positive at different Zn treatments of plant stages under drought stress except one case at anthesis (Table 3). Otherwise, under irrigation, before anthesis, the correlations between soluble proteins and soluble sugars were significantly negative at different Zn concentrations, except Zn = 250 ppm. The same correlations were true between soluble proteins and free amino acids. After anthesis, soluble proteins were negatively correlated with soluble sugars at Zn = 250 ppm and Zn-deficient treatments. Furthermore, the free amino acids had a significant negative correlation with both soluble proteins and soluble sugars at Zn = 250 ppm and Zn-deficient treatments.

Table 4 shows the effect of the presence or absence of Zn on the correlations between different metabolites in wheat cultivars. Commonly, there were positive correlations between metabolites in different cultivars at various plant stages under drought stress. However, under normal irrigation, a significant negative correlation existed between soluble proteins and free amino acids and soluble sugars in Sakha 69 and G186 before anthesis. The same negative correlation was detected between soluble proteins and soluble sugars in Sakha 69 and Gem 11 cultivars after anthesis.

## 4. Discussion

The obtained data indicated that the three metabolites, soluble sugars, soluble proteins, and free amino acids, were significantly affected by years (Table 5). The mean values of the metabolic compounds were larger in the first year than in the second year. This may have been due to change in climatic factors. Means of soluble sugars and proteins increased from the period before anthesis to that after anthesis in both years under drought stress and under normal irrigation for the four Zn levels. The concentrations of both metabolites were higher under drought stress than under irrigation, indicating that drought stress increased soluble sugars and soluble proteins. These results agree with those reported by [11,16,20]. In contrast, the authors of [6] found that, under moderate and severe drought stress, the content of soluble protein decreased while that of amino acids and proline increased. This may be due, when the plant exposed to drought, an osmotic balance occurs through the accumulation of soluble sugars, proline, and free amino acids, which increases the activity of enzymatic and nonenzymatic antioxidants.

Regarding the effect of Zn levels, the combined means of the concentration of soluble sugars were mostly the highest at Zn = 500 ppm under both environments, except at anthesis under normal irrigation, which ranked second (Table 2 and Figure 1). Furthermore, soluble sugars in cultivars were increased from 2 weeks before to 2 weeks after anthesis and gained higher contents under drought than under irrigation at the anthesis stage. Comparing the soluble sugar concentration of the control treatment (received no water or Zn = 0 ppm), it was higher under stress than under irrigation, proving that drought stress increased soluble sugars. It was noted that nano-ZnO alleviated drought-induced damage to the mitochondria and chloroplast, thus enhancing the drought tolerance [16,21,22]. At the Zn level of 500 ppm, the concentration of soluble sugars under both environments, except at anthesis under normal irrigation, proved the importance of foliar application of Zn to the soil under high pH (Table 1).

As in the case of soluble sugars, the effect of Zn levels on soluble proteins gradually increased from before to after anthesis (Figure 2). The soluble protein concentrations were higher under drought stress than under irrigation. The highest concentration of soluble proteins in the control and foliar application of water under drought stress or normal irrigation indicated an increase in soluble proteins caused by drought rather than Zn application. The authors of [20] concluded that Zn affects plant hormones, increases the stress proteins, and stimulates the antioxidant enzymes for alleviating drought effects, where sugar is measured according to [23] Furthermore, the authors of [24] indicated that the foliar application of Zn ameliorated the effects of severe drought stress by regulating the antioxidant defense mechanism and producing proline and soluble proteins. In contrast, the authors of [6] found that soluble proteins decreased while amino acids and proline increased under severe and moderate drought.

The results indicated that the free amino acids were increased before and after anthesis, reaching a higher concentration at anthesis under both irrigation regimes in the first than in the second year (Table 2 and Figure 3). Moreover, the amino acids were higher under normal irrigation than under drought stress. The effect of Zn levels showed no trend under drought stress or normal irrigation, indicating that the concentrations of amino acids were mainly affected by drought rather than Zn levels. The authors of [25] performed multiple regression analyses showing that 98% of genetic YDT (the ratio of yield in water-limited versus well-watered conditions) variance was explained by the drought responses of four (metabolites) amino acids (serine, asparagine, methionine, and lysine). The authors of [26] noted that the maximum decrease in total free amino acids was obtained under severe stress. In contrast, the authors of [27,28] found an increase in total free amino acids (proline, sugars, polyphenols, and glycine betaine) under drought stress.

The combined means of soluble sugars and protein concentration in the three cultivars over years and Zn levels increased from 2 weeks before anthesis to 2 weeks after anthesis (Figure 4 and Figure 5). The concentrations under drought stress were higher than under irrigation. The concentrations of both metabolites differed according to cultivar and plant stage, whether under drought or irrigation conditions [9].

Regarding total free amino acids, their concentrations were higher under normal irrigation than under drought stress at the three plant stages (Figure 6), reaching the maximum at anthesis, before decreasing sharply 2 weeks after anthesis, whether under irrigation or drought stress.

It was found that, at different Zn application levels, the most significant correlations between the soluble proteins and free amino acids in cultivars were positive under drought stress at the three plant stages. This indicated that the accumulation of amino acids led to soluble protein synthesis, which could be used as a criterion for evaluation of drought resistance among plants. In contrast, under normal irrigation, the soluble proteins were negatively correlated with soluble sugars before and after anthesis, but positively correlated at anthesis, indicating an adjustment mechanism in response to stress through active anabolism or a respiration sink. Likewise, the significant positive correlations were dominant between all investigated metabolites in different cultivars under the effect of Zn concentration levels under drought stress with some exceptions, such as soluble proteins and soluble sugars before and after anthesis, in the plants under irrigation.

## 5. Materials and Methods

In the 2018–2019 and 2019–2020 seasons, two experiments were carried at the Faculty of Agriculture Experimental Farm, Assiut University, Egypt (longitude: 31.125° N, latitude: 27.25° E, elevation: 45 m/148 ft). The first experiment was exposed to drought stress, and the second one was surface-irrigated as required, with a stripe of 6 m width in between both experiments to prevent water seepage. The experimental design was split-plot in RCBD with three replications. Foliar sprays of zinc oxide (500 ppm, 250 ppm), water, and control (no treatment) were assigned to the plot, and the three cultivars were allocated to the split plots. The plot had two rows, 3 m long and 30 cm apart. Seeds were sown on 28 November in the first and 27 November in the second season. After full emergence, seedlings were adjusted to 30 per row. Zinc oxide was applied via foliar spray two times: 2 weeks before anthesis and at anthesis.

### 5.1. The Genetic Materials

Three Egyptian wheat cultivars (*Triticum aestivum* L.), Sakha 69, Giza 68, and Gemmieza 11, were used.

Fe, Mn, Cu, and Zn levels were determined by an inductively coupled plasma emission spectrometer (iCAP 6200) in the central lab of the Faculty of Agriculture.

### 5.2. Irrigation

The drought stress experiment received planting irrigation and only one irrigation 3 weeks later. The second experiment received planting irrigation and four surface irrigations throughout the growing season.

### 5.3. Fertilization

During land preparation, super phosphate ($P_{2}O_{5}$, 15.5%) was added at a rate of 357.14 kg/ha, and nitrogen fertilization was added in the form of ammonium nitrate (33.5% N) to both experiments at a rate of 190.5 kg N/ha in one dose before the first irrigation.

The soil moisture percentage (Table 6) at 30 cm depth indicated that the plants in the drought stress experiment were subjected to severe drought starting before anthesis and lasting until harvest. It reached 23.86% in the first year and 22.81% in the second year before anthesis, which was less than the wilting point (28%) (Table 1).

### 5.4. Preparation of Plant Extracts for Analysis

To measure soluble metabolites, five flag leaves were sampled at 9:00 a.m. 2 weeks before anthesis, at anthesis, and 2 weeks after anthesis 3 days after foliar spraying of Zn. The leaves were placed in polyethene bags and transferred to the laboratory as quickly as possible to minimize water loss, washed rapidly in distilled water, and dried between filter paper layers. The excised leaves were then weighed (0.5 g), immediately crushed, and extracted in 10 mL of ice-cold distilled water [29]. The extracts were kept in a deep freezer until the time of analysis for soluble metabolites.

### 5.5. Determination of Water-Soluble Metabolites

Total soluble sugars were determined according to [23]. Total free amino acids were determined according to procedures described by [30]. Total soluble proteins were determined according to [31].

### 5.6. Statistical Analysis

Statistical analysis and mean separation using the LSD test (least significant difference) were performed on a plot mean basis according to [32] using Excel (Microsoft office 2016).
LSD (for Zn levels) = (2Ea/rb)0.05 × tά,
LSD (for cultivars) = (2Eb/ra)0.05 × tά
where Ea is the error a mean square, Eb is the error b mean square, r is the number of replications, and ά is the tabulated t-value at the degree of freedom of the proper error variance.

## 6. Conclusions

The three metabolites, soluble sugars, soluble proteins, and free amino acids were significantly affected by years. The mean values of the metabolic compounds were larger in the first year than in the second year, which may have been due to changes in climatic factors. Means of soluble sugars and proteins increased from the period before anthesis to that after anthesis in both years, and their concentrations were higher under drought stress than under irrigation, indicating that drought stress increased soluble sugars and soluble proteins. The concentration of soluble sugars increased with the increase in Zn levels at the three plant stages. In contrast, the concentration of soluble proteins was higher in the control treatment than at 500 and 250 ppm ZnO. The highest concentration of soluble proteins in the control and under the foliar application of water under drought or normal irrigation indicated an increase in soluble proteins caused by drought rather than Zn application. The concentration of free amino acids was higher under irrigation than under drought stress, reaching the maximum at anthesis, before decreasing sharply 2 weeks after anthesis, whether under irrigation or drought stress. In contrast, the zinc levels had no clear effect on free amino acids.

The ranking of cultivars for the three metabolites differed according to plant stage under both environments, reflecting their differential response to Zn and years.

It was found that, at different Zn application levels, the significant correlations between soluble proteins and free amino acid in cultivars were positive under drought stress at the three plant stages. This means that the accumulation of amino acids led to soluble protein synthesis, which can be used as the criterion for the evaluation of drought resistance in plants.

## Figures and Tables

**Figure 1 plants-11-01261-f001:**
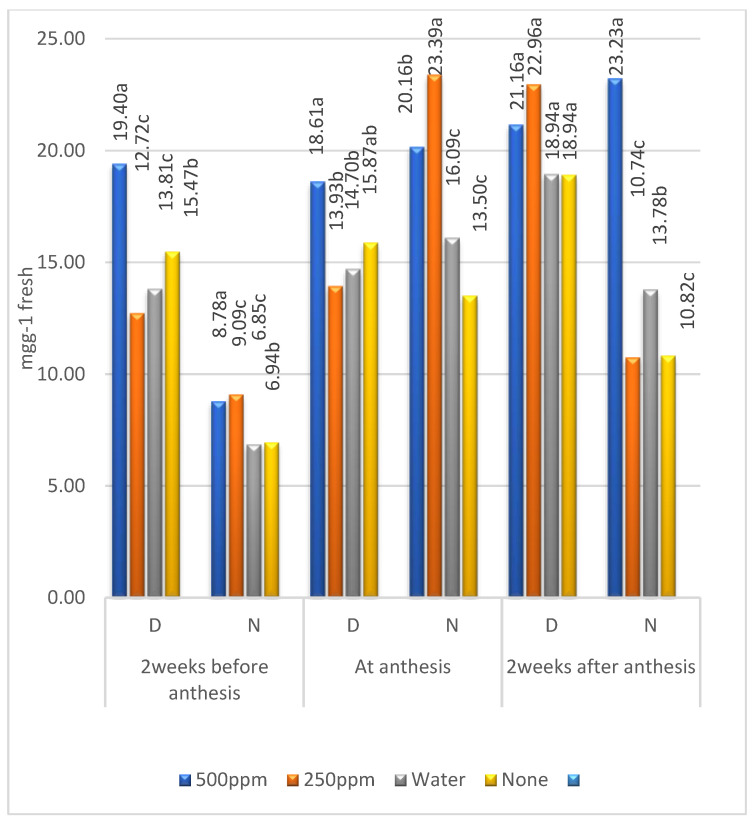
Combined means of soluble sugars for the levels of zinc under drought stress (D) and irrigation (N); means followed by the same letter in each environment are not significant at the 0.05 level of probability according to LSD test.

**Figure 2 plants-11-01261-f002:**
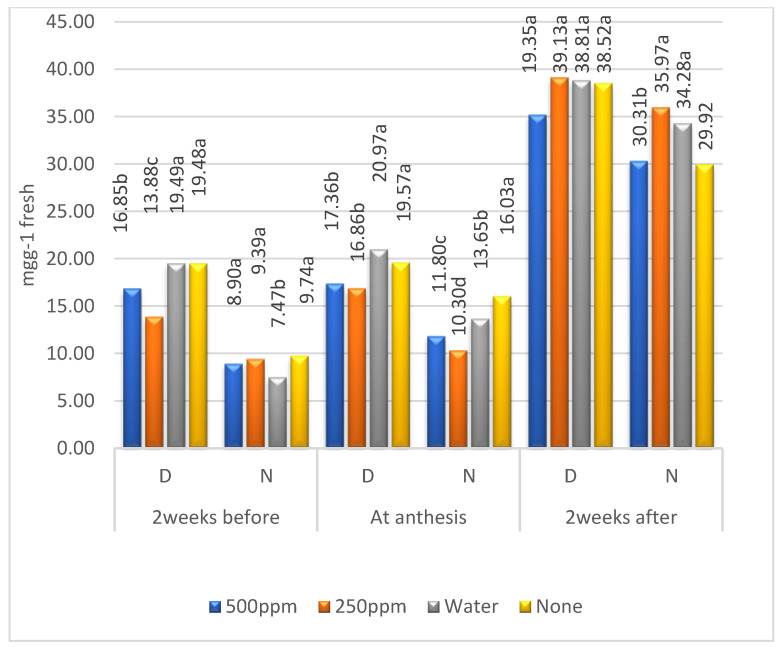
Combined means of soluble protein for the levels of zinc under drought stress (D) and irrigation (N); means followed by the same letter in each environment are not significant at the 0.05 level of probability according to LSD test.

**Figure 3 plants-11-01261-f003:**
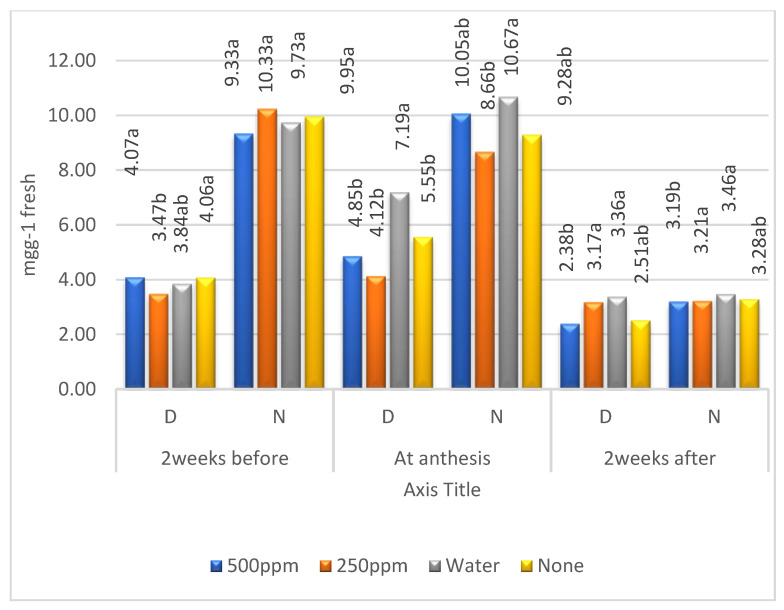
Combined means of free amino acids for the levels of zinc under drought stress (D) and irrigation (N); means followed by the same letter in each environment are not significant at the 0.05 level of probability according to LSD test.

**Figure 4 plants-11-01261-f004:**
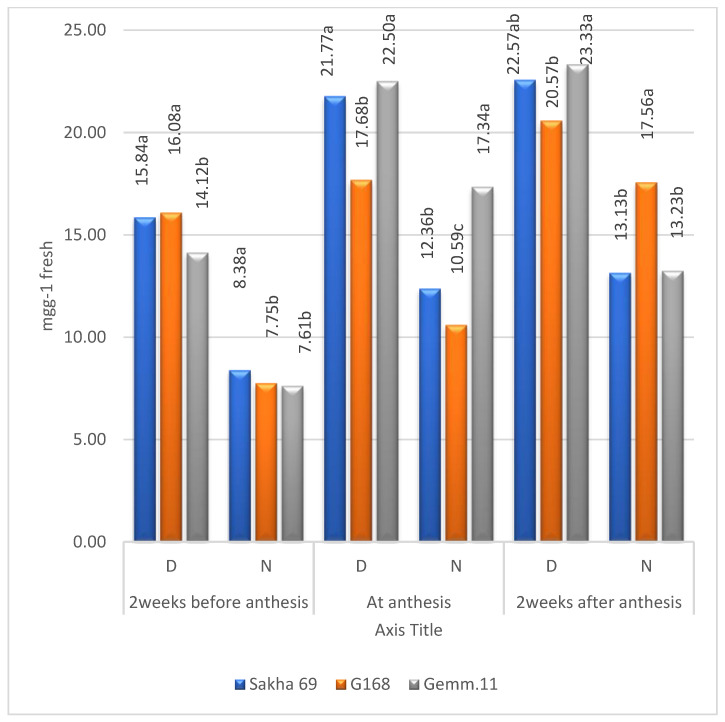
Combined means of soluble sugar for cultivars under drought stress (D) and irrigation (N); means followed by the same letter in each environment are not significant at the 0.05 level of probability according to LSD test.

**Figure 5 plants-11-01261-f005:**
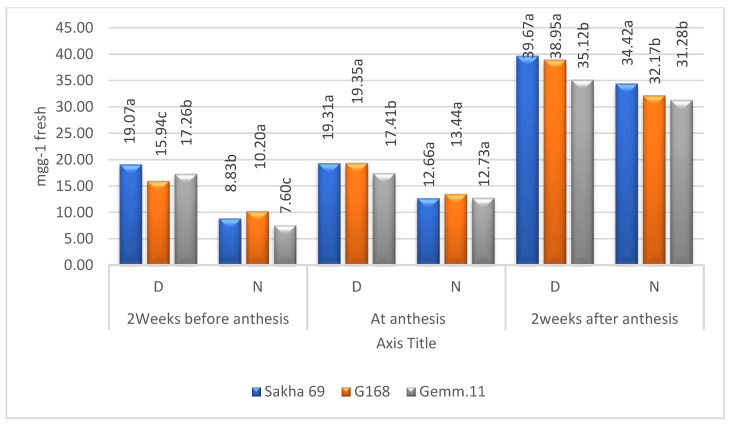
Combined means of soluble proteins for cultivars under drought stress (D) and irrigation (N); means followed by the same letter in each environment are not significant at the 0.05 level of probability according to LSD test.

**Figure 6 plants-11-01261-f006:**
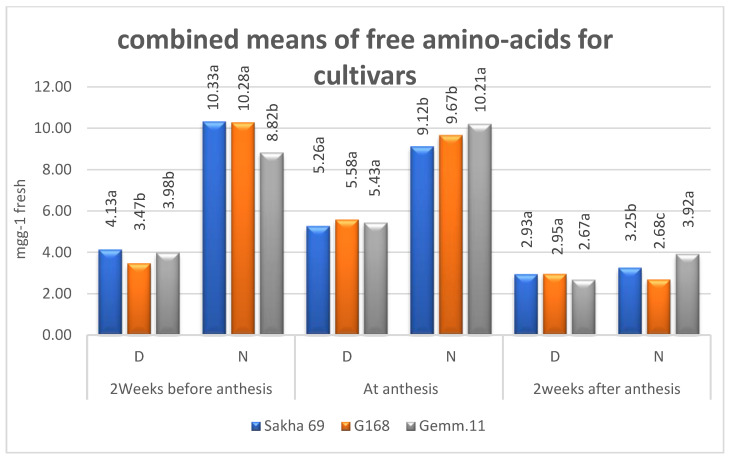
Combined means of free amino acids for cultivars under drought stress (D) and irrigation (N); means followed by the same letter in each environment are not significant at the 0.05 level of probability according to LSD test.

**Table 1 plants-11-01261-t001:** Mean squares of separate and combined analyses of the metabolites under drought stress and normal irrigation 2 weeks before anthesis (2wb), at anthesis, and 2 weeks after anthesis (2wa).

**Item**	**Soluble Sugars under Drought**			**Soluble Proteins under Drought**			**Free Amino Acids under Drought**		
	**Year 1**								
S.V.	2wb	Anthesis	2wa	2wb	Anthesis	2wa	2wb	Anthesis	2wa
Reps	11.30	1.050	6.230	6.290 *	49.58	15.00	1.130	4.640	0.600
Zn levels (A)	240.5 **	413.8 **	24.14	21.61 **	81.41 **	57.09 *	2.110	41.39 **	13.96 **
Error A	10.27	28.53	58.96	2.150	8.360	12.60	0.670	0.810	1.110
Cultivars (B)	128.3 **	384.1 **	297.4 **	12.77 *	29.42	158.3 **	2.690	0.150	4.940 **
A × B	61.45 **	181.3 *	539.6 **	33.16 **	10.46 **	130.3 **	3.100	12.63 **	3.710 **
Error B	8.400	61.14	40.36	2.260	8.420	13.70	1.470	1.110	0.790
	**Year 2**								
Reps	12.74	25.36	0.170	6.920	3.730	1.830	0.041	0.580	0.690
Zn levels (A)	12.74 **	200.7 **	49.30 **	145.0 **	40.30 **	300.3 *	0.520 **	2.460 *	3.280 **
Error A	0.860	0.080	2.860	3.000	3.240	53.39	0.055	0.400	0.140
Cultivars (B)	35.85 **	1.22	15.69 **	94.27 **	4.180	105.6 **	0.570	1.260 *	1.350
A × B	5.850 *	69.15 **	23.73 **	72.51 **	47.08 **	62.01 *	1.920 **	2.990 **	1.990 *
Error B	1.880	4.150	1.370	0.760	3.700	15.14	0.330	0.290	0.560
	**Combined**								
Years (Y)	1340 **	1874 **	1802 **	229.7 **	1203 **	42.96 **	363.7 **	197.9 **	47.57 **
Reps/years	7.260	13.21	3.200	6.610	26.66	8.410	0.590	2.610	0.640
Zn levels (A)	154.1 **	74.98 *	68.54	128.5 **	66.53 **	60.45	1.440 *	30.95 **	4.240 **
A × Y	99.14 **	539.5 **	4.890	38.08 **	55.18 **	297.0 **	1.200	12.90 **	13.00 **
Error A	5.56	14.3	30.91	2.580	5.800	32.99	0.360	0.610	0.630
Cultivars (B)	27.51 **	213.2 **	199.0 **	59.21 **	29.54	143.8 **	2.870	0.590	0.590
B × Y	136.6 **	172.1 *	114.1 **	47.83 **	5.060	120.2 **	0.380	0.810	5.710 **
A × B	47.33 **	82.03 *	363.88 **	45.44 **	105.0 **	118.42 **	3.85 **	11.30 **	5.350 **
A × B × Y	19.97 **	168.4 **	199.5 **	60.23 **	46.66 **	73.93 **	1.170	4.320 **	0.350
Error B	5.140	32.64	20.86	1.510	6.060	14.42	0.890	0.700	0.680
	**Soluble sugars under irrigation**			**Soluble proteins under irrigation**			**Free amino acids under irrigation**		
	**Year 1**								
S.V.	2wb	Anthesis	2waa	2wb	Anthesis	2wa	2wb	Anthesis	2w a
Reps	1.190	12.84	18.88	2.550	1.590	20.94	1.240	0.500	0.220
Zn levels (A)	13.67	946.6 **	823.2 **	32.87 **	250.0 **	447.75 **	18.00 **	5.800	3.400 **
Error A	2.890	23.04	9.090	2.690	2.300	8.950	1.420	1.460	0.290
Cultivars (B)	29.30 **	1226.4 **	344.0 **	88.67 **	7.980	65.79 *	5.510 *	21.75 **	11.96 **
A × B	11.11 **	1656 **	342.4 **	87.82 **	37.10 **	485.9 **	2.930	19.62 **	4.500 **
Error B	0.820	50.25	7.840	3.560	5.800	13.58	1.450	0.620	0.820
	**Year 2**								
Reps	0.230	0.040	0.300	0.180	0.290	2.980	1.560	0.430	1.410
Zn levels (A)	53.67 **	44.46 **	67.70 **	0.660 **	15.65 **	14.74	13.10 *	9.680	2.370
Error A	0.930	1.610	0.490	0.050	0.120	5.250	1.680	3.580	0.500
Cultivars (B)	442.9 **	129.7 **	17.83 *	0.740 *	1.740	46.69 *	36.50 **	33.59 **	1.320
A × B	14.66 **	25.19 **	38.35 **	2.020 **	1.720 **	22.09 *	11.45 **	22.80 **	0.630
Error B	0.580	2.890	2.930	0.120	0.050	8.080	1.330	2.050	0.680
	**Combined**								
Years (Y)	413.1 **	5930 **	216.3 **	4148 **	6459 **	4729 **	145.4 **	767.7 **	10.98 *
Reps/years	0.71	6.44	9.59	1.37	0.94	11.96	1.400	0.460	0.820
Zn levels (A)	25.22 **	344.5 **	625.7 **	17.86 **	109.63 **	159.6 **	2.610	13.85 *	0.280
A × Y	42.13 **	646.5 **	265.2 **	15.67 **	156.02 **	302.9 **	28.50 **	1.640	5.500 **
Error A	1.910	12.32	4.790	1.370	1.210	7.100	1.550	2.520	0.390
Cultivars (B)	3.970 **	1070 **	153.69 **	40.53 **	4.440	62.66 **	17.55 **	7.160 **	9.170 **
B × Y	68.27 **	286.3 **	208.1 **	48.89 **	5.290	49.81 *	24.45 **	48.18 **	4.110 **
A × B	14.77 **	821.41 **	147.74 **	47.79 **	26.59 **	309.2 **	7.650 **	21.79 **	2.790 **
A × B × Y	11.00 **	860.2 **	233.0 **	42.06 **	12.23 *	198.8 **	6.730 **	20.63 **	2.340
Error B	0.700	26.57	5.390	1.840	2.930	10.83	1.390	1.340	0.750

* Significant at 0.05 level of probability; ** significant at 0.01 level of probability.

**Table 2 plants-11-01261-t002:** Means of the studied traits under drought stress and normal irrigation in the 2 years and combined means over cultivars.

**Soluble Sugar under Drought, mg·g^−1^ Fresh Weight**									
**Zn Levels**	**2 Weeks before Anthesis**			**at Anthesis**			**2 Weeks after at Anthesis**		
	**Y1**	**Y2**	**Comb**	**Y1**	**Y2**	**Comb**	**Y1**	**Y2**	**Comb**
500 ppm	26.50a	12.29a	19.40a	29.01a	8.210c	18.61a	26.59a	15.72b	21.16a
250 ppm	14.08b	11.36a	12.72c	21.61b	6.240d	13.93b	27.20a	18.71a	22.96a
Water	18.19bc	9.430b	13.81c	12.44c	17.07a	14.70b	24.05a	13.84b	18.94a
None	19.88c	11.06a	15.47b	20.51b	11.24b	15.87ab	24.14a	13.68b	18.91a
**Soluble sugar under irrigation, mg·g^−1^ fresh weight**									
**Zn levels**	**2 Weeks before anthesis**			**at anthesis**			**2 weeks after at anthesis**		
	**Y1**	**Y2**	**Comb**	**Y1**	**Y2**	**Comb**	**Y1**	**Y2**	**Comb**
500 ppm	10.72b	6.870a	8.780a	34.11a	6.22c	20.16b	27.14a	19.32a	23.23a
250 ppm	13.42a	4.740b	9.090a	37.51a	9.280b	23.39a	8.200b	13.28d	10.74c
Water	9.470c	4.220b	6.850b	22.40b	9.790b	16.09c	9.630b	17.92b	13.78b
None	7.600d	6.270b	6.940b	15.43c	11.56a	13.50c	6.670b	14.98c	10.82c
**Soluble protein under drought, mg·g^−1^ fresh weight**									
**Zn levels**	**2 Weeks before anthesis**			**at anthesis**			**2 weeks after at anthesis**		
	**Y1**	**Y2**	**Comb**	**Y1**	**Y2**	**Comb**	**Y1**	**Y2**	**Comb**
500 ppm	18.28ab	15.42ab	16.85b	23.12a	11.60b	17.36b	48.58a	21.80a	35.19a
250 ppm	17.66b	10.09c	13.88c	18.96b	14.76a	16.86b	46.45a	31.80a	39.13a
Water	19.80a	19.19a	19.49a	26.30a	15.64a	20.97a	42.60b	35.02a	38.81a
None	21.10a	17.84a	19.48a	22.72a	16.42a	19.57a	44.91ab	32.13a	38.52a
**Soluble protein under irrigation, mg·g^−1^ fresh weight**									
**Zn levels**	**2 Weeks before anthesis**			**At anthesis**			**2 weeks after at anthesis**		
	**Y1**	**Y2**	**Comb**	**Y1**	**Y2**	**Comb**	**Y1**	**Y2**	**Comb**
500 ppm	16.45a	1.360a	8.900a	19.18c	4.410a	11.80c	34.36b	26.26a	30.31b
250 ppm	17.14a	1.640a	9.390a	18.16c	2.450b	10.30c	48.55a	23.39a	35.97a
Water	13.87b	1.080a	7.470b	22.50b	4.790a	13.650b	44.82a	23.75a	34.28a
None	18.40a	1.070a	9.740a	29.82a	2.230b	16.020a	35.18b	24.67a	29.92b
**Free amino acids under drought, mg·g^−1^ fresh weight**									
**Zn levels**	**2 Weeks before anthesis**			**at anthesis**			**2 weeks after at anthesis**		
	**Y1**	**Y2**	**Comb**	**Y1**	**Y2**	**Comb**	**Y1**	**Y2**	**Comb**
500 ppm	6.630a	1.520b	4.070a	6.410b	3.280b	4.840c	3.090ab	1.670c	2.380b
250 ppm	5.460a	1.480b	3.470b	4.820c	3.420b	4.120c	4.860a	1.470c	3.170a
Water	6.200a	1.480b	3.840a	9.950a	4.430a	7.190a	4.520a	2.210b	3.360a
None	6.150a	1.970a	4.060a	7.160b	3.940ab	5.550b	2.200b	2.820a	2.500b
**Free amino acids under irrigation, mg·g^−1^ fresh weight**									
**Zn levels**	**2 Weeks before anthesis**			**at anthesis**			**2 weeks after at anthesis**		
	**Y1**	**Y2**	**Comb**	**Y1**	**Y2**	**Comb**	**Y1**	**Y2**	**Comb**
500 ppm	10.26bc	8.390a	9.330a	13.44a	6.670a	10.05ab	3.380b	2.990ab	3.190a
250 ppm	10.49b	9.970a	10.23a	11.52a	5.800a	8.660b	3.990ab	2.440b	3.210a
Water	12.93a	6.540b	9.73a	13.92a	7.420a	10.67a	4.350a	2.570b	3.460a
None	11.24ab	8.660a	9.95a	12.85a	5.710a	9.280ab	2.980b	3.580a	3.280a

Means followed by the same letter in each column are not significantly different at the 0.05 level of probability, according to the LSD test. Y1 = the first year, Y2 = the second year.

**Table 3 plants-11-01261-t003:** The correlations among soluble sugars (SS), soluble proteins (SP), and free amino acids (AA) for Zn levels under drought (above) and irrigation (below diagonal) at different plant stages.

Zn Levels	Plant Stages	2 Weeks before Anthesis			at Anthesis			2 Weeks after Anthesis		
	Compound	SS	SP	AA	SS	SP	AA	SS	SP	AA
500 ppm	SS		0.94 **	0.97 **		0.98 **	0.87 *		0.96 **	0.82 *
	SP	−0.99 **		0.86 *	0.98 **		0.93 **	0.99 **		0.79
	AA	1.00 **	−1.00 **		0.98 **	0.97 **		0.41	0.31	
250 ppm	SS		0.74	0.56		0.92 **	0.93 **		0.71	0.65
	SP	−0.50		0.97 **	0.99 **		0.89 *	−0.99 **		1.00 **
	AA	0.87 *	−0.53		0.98 **	0.97 **		0.93 **	−0.89 *	
water	SS		0.16	1.00 **		−0.73	−0.82 *		0.65	0.70
	SP	−0.91 **		0.15	0.98 **		0.99 **	−0.98 **		0.51
	AA	0.97 **	−0.96 **		0.95 **	0.99 **		0.96 **	−0.90 *	
none	SS		0.99 **	0.98 **		0.98 **	0.90 *		0.94 **	−0.82 *
	SP	−0.87 *		0.96 **	0.61		0.83 *	−0.97 **		−0.84 *
	AA	0.96 **	−0.90 *		0.41	0.97 **		−0.72	0.74	

*, ** Significant at the 0.05 and 0.01 probability levels, respectively.

**Table 4 plants-11-01261-t004:** The correlations among soluble sugars (SS), soluble proteins (SP) and free amino acids (AA) for the cultivars under drought (above) and irrigation (below diagonal) at different plant stages.

Cultivars	Plant Stages	2 Weeks before Anthesis			at Anthesis			2 Weeks after Anthesis		
	Compound	SS	SP	AA	SS	SP	AA	SS	SP	AA
Sakha 69	SS		0.17	0.71		0.68	0.78		0.97 **	0.96 **
	SP	−0.96 **		0.24	0.97 **		0.93 **	−0.87 *		0.88 *
	AA	0.99 **	−0.97 **		0.96 **	1.00 **		0.50	−0.74	
G168	SS		0.94 **	−0.02		0.89 *	0.93 **		0.90 *	0.79
	SP	−0.98 **		−0.04	0.80 *		0.99 **	0.66		0.91 *
	AA	0.98 **	−0.99 **		0.83 *	0.99 **		0.49	0.34	
Gem 11	SS		0.79	0.97 **		0.96 **	0.99 **		0.98 **	0.59
	SP	−0.68		0.70	1.00 **		0.94 **	−1.00 **		0.61
	AA	0.98 **	−0.56		0.98 **	0.98 **		0.86*	−0.87 *	

*, ** Significant at the 0.05 and 0.01 probability levels, respectively.

**Table 5 plants-11-01261-t005:** Some physical and chemical properties of representative soil sample in the experimental sites before sowing (30 cm depth).

Item	Value	Item	Value
Sand (%)	27.4	Total nitrogen (%)	0.72
Silt (%)	24.3	KCl-extractable N (mg·kg^−1^)	41.23
Clay (%)	48.3	Fe mg/kg	13.21
Texture grade	Clay	Mn mg/kg	5.152
EC (1:1 extract) dS·m^−1^	0.47	Cu mg/kg	1.31
pH	8.2	Zn mg/kg	2.12
CaCO_3_ (%)	3.4	Soil moisture at F. capacity	46%
Organic matter (%)	1.75	Soil moisture at wilting point.	28%
NaHCO_3_-extractable P (mg·kg^−1^)	4.36	NH_4_OAC-extractable K (mg·kg^−1^)	49.24

Each value represents the mean of three replications.

**Table 6 plants-11-01261-t006:** The soil moisture percentage at 30 cm depth, estimated from a mixed sample representing different areas of the experimental site.

Seasons		Season 2018/19		Season 2019/20	
	Experiment	Drought Stress Experiment	Normal Irrig. Experiment	Drought Stress Experiment	Normal Irrig. Experiment
Time					
Before 2nd irrigation		37.84	39.84	38.56	37.12
Before 4th irrigation		23.86	33.72	22.81	36.89
At anthesis		17.15	38.17	18.52	37.55

## Data Availability

All data are included in the manuscript.

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
