# Peer review of "Changes in Carbon and Nitrogen Metabolites before, at, and after Anthesis for Wheat Cultivars in Response to Reduced Soil Water and Zinc Foliar Application"

_plants, 2022, doi:10.3390/plants11091261_

Round 1

Reviewer 1 Report

The manuscript studied the changes of carbon and nitrogen metabolites in wheat for drought and zinc dificiency stress. The research is meaningful and the writting is well. However, some items should be carefully revised.

  1. All the data must be expressed as means ± SD in the tables and manuscript.
  2. The data should be expressed as 4 significant digits, like 6.29 change to 6.290, 240.50 change to 240.5, and 1202.9 change to 1203.
  3. For table 1, what the means of "*, **; significant at 0.05."? Please rewrite it more accurate.
  4. And also, the significant difference should be used for table 1 data.
  5. From figure 1 to figure 6, I have not found the standard deviation, how to caculate the significant difference?
  6. More details of methods must be provided in the manuscript.

Author Response

First Reviewer

Dear sir

First of all, thank you for this good review

I would like to denote that the purpose of using SE with the mean is to know the significancy of the mean. If the mean ≥ the SE by 1.96 then the mean is significant at 0.05% level of probability and we symbolize it by (*), and If the mean ≥ the SE by 2.57 the mean is significant at 0.01% level of probability, denoting it by (**).

So, using (*) with mean squares values make no need to follow them by SE.

SD is the dispersion of observations around population mean, while SE is the dispersion of samples means around population mean. This work is considered as a sample from a population, so we do not use SD.

  • All the data must be expressed as means + SD in the tables and manuscript.
  • In table 1 includes the mean squares of the split-plot design followed by significant signs (* or **). Thus, any mean square should not be followed by SE or SD.
  • Table 2 includes the means; each mean followed by an alphabetical letter pointing to significant differences between treatments based on LSD test.
  • Tables 3 and 4 display correlation values followed by significant signs (* or **).
  • In tables 5 and 6 the estimation of physical and chemical properties and soil moisture content was done on a mixed sample representing different areas of the experimental site, so there is no need to SE or SD.
  • In all figures, each bar headed by an alphabetical letter denotes significant differences between treatments based on LSD test.
  • The data should be expressed as 4 significant digits, like 6.29 change to 6.290, 240.50 change to 240.5, and 1202.9 change to 1203

     The data was edited.

  • For table 1, what the means of "* **; significant at 0.05."? Please rewrite it more accurate

      These refer to the significance of Zn   levels, cultivars and different interactions.

  • And also, the significant difference should be used for table 1 data.

     Was explained in comment (1)

  • From figure 1 to figure 6, I have not found the standard deviation, how to calculate the significant difference?

     Was explained in comment (1)

  • More details of methods must be provided in the manuscript

      Was edited in statistical analysis and captions of the tables in the newly attached manuscript file, or kindly refer to the undetailed part of the methods.

Thanks a lot for the valuable comments.

Reviewer 2 Report

First of all, I want to thank all authors for this interesting manuscript.

Mahdy et al., study the effect of foliar application of Zn oxide on soluble sugar, proteins, amino acids, and alleviation of drought stress on wheat (Triticum aestivum L.) under normal irrigation and drought stress environments, along with studying the correlations between various primary metabolites of wheat cultivars under the effects of Zn application. Studying the reproduction stage, to make the biochemical measurement plant that before anthesis, at anthesis, after anthesis is a very taught work.

All my comments are minor to enhance the overall quality of the paper to deserve to be published in the Q1 journal (Plants)

1- Give hint in the title and in the last paragraph of the intro that ur hypothesis regarding the comparison between - before anthesis, at anthesis, after anthesis.

2- All figures need to be more professional and informative with SE or SD - with post hoc test -LSD- Duncan or tuky with letter of significant

3- Table1 and 2 are hard to follow its detail, I suggest converting to figures and making a small table for the combined effect in supplementary data.

4- Also I suggest drawing a small summary figure to ease the understanding of the discussion.

5- Attached file for the grammar and tempo errors.

Good luck

Minor revision

Author Response

Changes in Carbon and nitrogen metabolites before, at and after anthesis for wheat cultivars in response to reduced soil water and zinc foliar application

This is the new title as needed in the first comment.

Please attach the word file for accepting all the changes.

The attached file represents the  First and second reviewers edits.

Reviewer 3 Report

In this manuscript, the authors studied the changes in Carbon and nitrogen metabolites in three wheat cultivars in response to reduced soil water and zinc foliar application. In this study authors conducted two field experiments were conducted in the 2018/19 and 2019/20 seasons to explore the effectiveness of the foliar application of zinc oxide on soluble sugar, soluble proteins, and free amino acids under normal irrigation and drought-stressed environments. Three Egyptian wheat cultivars (Triticum atrium L) were used. The experimental design was a split-plot in RCBD with three replications, applying Zinc oxide levels to the whole plot and the split plots. Leaf samples were taken for analysis before anthesis, at anthesis, and after anthesis. Application of Zn increased soluble sugars. However, the free-amino acids were higher under irrigation, reached their maximum at anthesis, and decreased sharply after two weeks from anthesis. Cultivars' rank for the three metabolites differed from one plant stage to another reflecting their response to Zn and years. Correlations between metabolites according to Zn were positive. The findings suggest the foliar application of Zn to alleviate drought stress. The manuscript is very well written and I found no major drawbacks in the manuscript and hence can be accepted after some minor changes in the scientific writing.

Author Response

Thanks a lot for your kind reply

Round 2

Reviewer 1 Report

The revised manuscript could be accepted  for publication.